# A Data-Free Approach to Mitigate Catastrophic Forgetting in Federated Class Incremental Learning for Vision Tasks

**Sara Babakniya**
Computer Science
University of Southern California
Los Angeles, CA
babakniy@usc.edu

**Zalan Fabian**
Electrical and Computer Engineering
University of Southern California
Los Angeles, CA
zfabian@usc.edu

**Chaoyang He**
FedML
Sunnyvale, CA
ch@fedml.ai

**Mahdi Soltanolkotabi**
Electrical and Computer Engineering
University of Southern California
Los Angeles, CA
soltanol@usc.edu

**Salman Avestimehr**
Electrical and Computer Engineering
University of Southern California
Los Angeles, CA
avestime@usc.edu

## Abstract

Deep learning models often suffer from forgetting previously learned information when trained on new data. This problem is exacerbated in federated learning (FL), where the data is distributed and can change independently for each user. Many solutions are proposed to resolve this catastrophic forgetting in a centralized setting. However, they do not apply directly to FL because of its unique complexities, such as privacy concerns and resource limitations. To overcome these challenges, this paper presents a framework for **federated class incremental learning** that utilizes a generative model to synthesize samples from past distributions. This data can be later exploited alongside the training data to mitigate catastrophic forgetting. To preserve privacy, the generative model is trained on the server using data-free methods at the end of each task without requesting data from clients. Moreover, our solution does not demand the users to store old data or models, which gives them the freedom to join/leave the training at any time. Additionally, we introduce SuperImageNet, a new regrouping of the ImageNet dataset specifically tailored for federated continual learning. We demonstrate significant improvements compared to existing baselines through extensive experiments on multiple datasets.

## 1 Introduction

Federated learning (FL) [40, 29] is a decentralized machine learning technique that enables privacy-preserving collaborative learning. In FL, multiple users (clients) train a common (global) model in coordination with a server without sharing personal data. In recent years, FL has attracted tremendous attention in both research and industry and has been successfully employed in various fields, such as autonomous driving [17], next-word prediction [21], health care [13], and many more.

Despite its popularity, deploying FL in practice requires addressing critical challenges, such as resource limitation and statistical and system heterogeneity [27, 33]. While tackling these challenges is an essential step towards practical and efficient FL, there are still common assumptions in most FL frameworks that are too restrictive in realistic scenarios.

37th Conference on Neural Information Processing Systems (NeurIPS 2023).

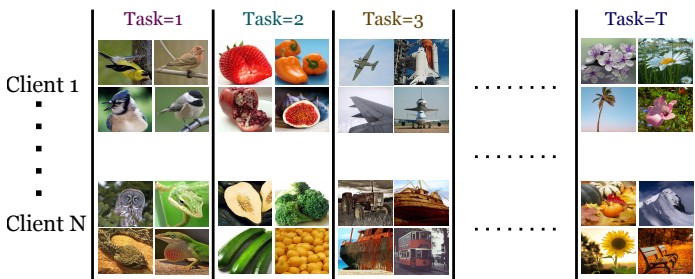

Figure 1: In the real world, users constantly change their interests, observe new data, or lose some of the old ones. As a result, the training dataset is divided into different tasks. For example, here, at $Task = 1$, the clients' datasets dominantly include pictures of animals, and by the end of the training ($Task = T$), the trend shifts towards landscapes.

In particular, one of the most common assumptions is that clients' local data distribution is fixed and does not change over time. However, in real-world applications [49], clients' data constantly evolve due to changes in the environment, trends, or new interests. For example, [6] presents the real-world data of an online shop, suggesting interest in items shifts through seasons. Another example arises in healthcare, where a model trained on old diseases should be able to generalize to new diseases [58]. In such scenarios (Figure 1), the model must rapidly adapt to the incoming data while preserving performance on past data distributions to avoid catastrophic forgetting [28, 39].

In the centralized setting, such problems have been explored in continual learning [48, 34] (also called lifelong learning [3] or incremental learning [9, 7] based on the initial settings and assumptions). In recent years, various algorithms have been proposed in Continual Learning (CL) to tackle catastrophic forgetting from different angles and can achieve promising performance in different scenarios.

Despite all the significant progress, most CL methods are not directly applicable to the federated setting due to inherent differences (Table 1) between the two settings. For instance, experience replay [47] is a popular approach, where a portion of past data points is saved to maintain some representation of previous distributions throughout the training. However, deploying experience replay in FL has resource and privacy limitations. It requires clients to store and keep their data, which may increase the memory usage of already resource-limited clients. Furthermore, users may not be able to store data for more than a specific time due to privacy concerns. Finally, depending solely on the clients to preserve the past is not reliable, as clients leaving means losing their data.

| Challenge | Limitation |
|---|---|
| Low memory | Clients cannot store many examples |
| Clients drop out | Causes loss of information stored in memory |
| New clients join | New clients only have access to new classes |
| Privacy | Limits data saving and sharing of the clients |

Table 1: Challenges that limit the direct use of continual learning methods in federated settings.

To address the aforementioned problems, we propose MFCL, *Mimicking Federated Continual Learning*: a privacy-preserving federated continual learning approach without episodic memory. In particular, MFCL is based on training a generative model in the server and sharing it with clients to sample synthetic examples of past data instead of storing the actual data on the client side. The generative model training is data-free in the sense that no form of training data is required from the clients, and only the global model is used in this step. It is specifically crucial because this step does not require powerful clients and does not cause any extra data leakage. Finally, this algorithm has competitive performance; our numerical experiments demonstrate improvement by $10\% - 20\%$ in average accuracy while reducing the training overhead of the clients.

Moreover, benchmarking federated continual learning in practical scenarios requires a large dataset to split among tasks and clients. However, existing datasets are not sufficiently large, causing most of the existing works in federated continual learning evaluating on a few clients (5 to 20) [45, 24, 52]. To enable more practical evaluations, we release a new regrouping of the ImageNet dataset, *SuperImageNet*. SuperImageNet enables evaluation with many clients and ensures all clients are assigned sufficient training samples regardless of the total number of tasks and active clients.

We summarize our contributions below:

- We propose a novel framework to tackle the federated class incremental learning problem more efficiently for many users. Our framework specifically targets applications where past data samples on clients are unavailable.
- We point out potential issues with relying on client-side memory for FCL. Furthermore, we propose using a generative model trained by the server in a *data-free manner* to help overcome catastrophic forgetting while preserving privacy.
- We modify the client-side training of traditional FL techniques in order to mitigate catastrophic forgetting using a generative model.
- We propose a new regrouping of the ImageNet dataset, SuperImageNet, tailored to federated continual learning settings that can be scaled to a large number of clients and tasks.
- We demonstrate the efficacy of our method in more realistic scenarios with a larger number of clients and more challenging datasets such as CIFAR-100 and TinyImageNet.

## 2   Related Work

**Continual Learning.** Catastrophic forgetting [39] is a fundamental problem in machine learning: when we train a model on new examples, its performance degrades when evaluated on past data. This problem is investigated in continual learning (CL) [59], and the goal is for the model to learn new information while preserving its knowledge of old data. A large body of research has attempted to tackle this problem from different angles, such as adding regularization terms [31, 1, 41], experience replay by storing data in memory [2, 10, 4, 35], training a generative model [56, 53, 32], or architecture parameter isolation [16, 38, 19, 51].

In CL settings, the training data is presented to the learner as a sequence of datasets - commonly known as **tasks**. In each timestamp, only one dataset (task) is available, and the learner's goal is to perform well on all the current and previous tasks.

Recent work focuses on three main scenarios, namely task-, domain- and class-incremental learning (IL) [54]. In *Task-IL*, tasks are disjoint, and the output spaces are separated by task IDs provided during training and test time. For *Domain-IL*, the output space does not change for different tasks, but the task IDs are no longer provided. Finally, in *Class-IL*, new tasks introduce new classes to the output space, and the number of classes increases incrementally. Here, we work on **Class-IL**, which is the more challenging and realistic, especially in FL. In most of the FL applications, there is no task ID available, and it is preferred to learn a *single* model for all the observed data.

**Class Incremental Learning.** In standard centralized Class-IL, the model is trained on a sequence of non-overlapping $T$ tasks $\{\mathcal{T}^{(1)}, \mathcal{T}^{(2)}, ..., \mathcal{T}^{(T)}\}$ where the data distribution of task $t$, $D^t$, is fixed but unknown in advance, while all the tasks share the same output space ($\mathcal{Y}$). For task $t$, $D^t$ consists of $N^t$ pairs of samples and their labels $\{(x_i^t, y_i^t)\}_{i=1}^{N^t}\}$, where all the newly introduced classes ($y_i^t$) belong to $\mathcal{Y}^t$ ($y_i^t \in \{\mathcal{Y}^t\}$ and $\bigcup_{j=1}^{t-1}\{\mathcal{Y}^j\}\bigcap\{\mathcal{Y}^t\} = \emptyset$). Moreover, a shared output space among all tasks means that at the end of task $t$, the total number of available classes equals $q = \sum_{i=1}^t |\mathcal{Y}^i|$.

**Federated Continual Learning.** In real-life scenarios, users' local data is not static and may evolve. For instance, users' interests may change over time due to seasonal variations, resulting in more examples for a given class. On the other hand, reliability issues or privacy concerns may lead to users losing part of their old data as well. In Federated Continual Learning (FCL), the main focus is to adapt the global model to new data while maintaining the knowledge of the past.

Even though FCL is an important problem, it has only gained attention very recently, and [58] is the first paper on this topic. It focuses on Task-IL, which requires a unique task ID per task during inference. Furthermore, it adapts separate masks per task to improve personalized performance without preserving a common global model. This setting is considerably different than ours as we target Class-IL with a single global model to classify all the classes seen so far. [37] employs server and client-side knowledge distillation using a surrogate dataset. [15] relaxes the problem as clients have access to large memory to save the old examples and share their data, which is different from the standard FL setting. Some works, such as [26, 44, 52], explore the FCL problem in domains other than image classification. [42] has proposed using variational embedding to send data to the server securely and then server-side training to rehearse the previous task for Domain-IL.

This work focuses on Class-IL for supervised image classification without memory replay, similar to [45, 24]. However, [24] allows overlapping classes between tasks and focuses on few-shot learning, which is different from the standard Class-IL. The most related work to ours is [45], where authors propose FedCIL. This work also benefits from generative replay to compensate for the absence of old data and overcome forgetting. In FedCIL, clients train the discriminator and generator locally. Then, the server takes a consolidation step after aggregating the updates. In this step, the server generates synthetic data using all the generative models trained by the clients to consolidate the global model and improve the performance. The main difference between this work and ours is that in our work, the generative model is trained by the server in a data-free manner, which can reduce clients' training time and computation and does not require their private data (detailed comparison in Appendix H).

**Data-Free Knowledge Distillation.** Knowledge distillation (KD) [25] is a popular method to transfer knowledge from a well-trained teacher model to a (usually) smaller student model. Common KD methods are data-driven, and at least a small portion of training data is required. However, in some cases, training data may not be available during knowledge distillation due to privacy concerns.

To tackle this problem, a new line of work [12, 22] proposes *data-free knowledge distillation*. In such methods, a generative model is used as a training data substitute. This generative model is trained to generate synthetic images such that the teacher model predicts them as their assigned label (Figure 2). This method has recently become popular in CL [57, 50] as well, mainly due to the fact that it can eliminate the need for memory in preserving knowledge. Data-free KD has been previously used in FL [60] to reduce the effect of data heterogeneity. However, to the best of our knowledge, this is the first work that adapted such a technique in the context of federated continual learning.

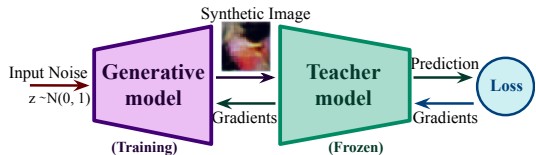

Figure 2: Data-Free Knowledge Distillation. The generator receives random noise as input labels and synthesizes images that are labeled correctly by the trained teacher model.

## 3   Federated Class Incremental Learning with MFCL

In federated Class-IL, a shared model is trained on $T$ different tasks. However, the distributed and private nature of FL makes it distinct from the centralized version. In FL, users may join, drop out, or change their data independently. Besides, required data or computation power for some centralized algorithms may not be available in FL due to privacy and resource constraints.

To address the aforementioned problems, we propose MFCL, which is less reliant on the client-side memory and computational power. This algorithm includes two essential parts: *first*, at the end of each task, the server trains a generative model with data-free knowledge distillation methods to learn the representation of the seen classes. *Second*, clients can reduce catastrophic forgetting by generating synthetic images from the trained generative model obtained from the server side. This way, clients are not required to use their memory for storing old data. Moreover, this technique can address the problem of newly connected clients without past data. Furthermore, since the server trains the generative model training without additional information, this step does not introduce **new** privacy issues. Finally, MFCL can help mitigate the data heterogeneity problem, as clients can synthesize samples from classes they do not own [60] in memory. Next, we explain the two key parts of MFCL: server-side generative model (Figure 3 Left) and client-side continual learning (Figure 3 Right).

### 3.1   Server-Side: Generative Model

The motivation for deploying a generative model is to synthesize images that mimic the old tasks and to avoid storing past data. However, training these generative models on the client's side, where the training data exists, is *computationally expensive*, *requires a large amount of training data* and can be potentially *privacy concerning*. On the other hand, the server has only access to the global model and aggregated weights and no data. We propose training a generative model on the server, but in a data-free manner, i.e., utilizing model-inversion image synthesis [57, 50]. In such approaches, the goal is to synthesize images optimized with respect to the discriminator (global model). Then, the

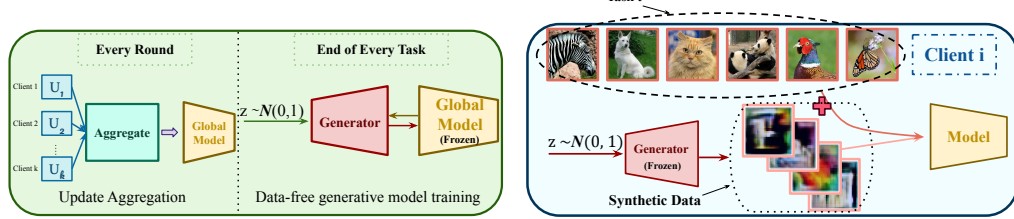

Figure 3: Overview of MFCL. **Left.** The server aggregates the updates every round and trains a generator using data-free methods at the end of each task. **Right.** Clients train their models locally using their local data and synthetic images of past tasks from the generator.

generative model is shared with the clients to generate images during local training. To this aim, we utilize a generative model with ConvNet architecture, $\mathcal{G}$, that takes noise $z \sim \mathcal{N}(0,1)$ as input and produces a synthetic sample $\tilde{x}$, resembling the original training input with the same dimensions. In order to train this model, we must balance the various training objectives we detail next.

**Cross Entropy Loss.** First, the synthetic data should be labeled correctly by the current discriminator model (global model or $\mathcal{F}$). To this end, we employ cross entropy classification loss between its assigned label $z$ and the prediction of $\mathcal{F}$ on synthetic data $\tilde{x}$. Note that noise dimension can be arbitrary and greater than the current discovered classes of task $t$; therefore, we only consider the first $q$ dimension here, where $q = \sum_{i=1}^{t} |\mathcal{Y}^i|$ (which is equal to the total number of classes seen in the previous tasks). Then, we can define the cross-entropy loss as

$$\mathcal{L}_{CE} = CE(argmax(z[:q]), \mathcal{F}(\tilde{x})). \tag{1}$$

**Diversity Loss.** Synthetic images can suffer from a lack of class diversity. To solve this problem, we utilize the information entropy (IE) loss [12]. For a probability vector $\mathbf{p} = (p_1, p_2, ..., p_q)$, information entropy is evaluated as $\mathcal{H}_{info}(\mathbf{p}) = -\frac{1}{q} \sum_i p_i \log(p_i)$. Based on the definition, inputs with uniform data distributions have the maximum IE. Hence, to encourage $\mathcal{G}$ to produce diverse samples, we deploy the diversity loss defined as

$$\mathcal{L}_{div} = -\mathcal{H}_{info}(\frac{1}{bs} \sum_{i=1}^{bs} \mathcal{F}(\tilde{x}_i)). \tag{2}$$

This loss measures the IE for samples of a batch ($bs$: batch size). Maximizing this term encourages the output distribution of the generator to be more uniform and balanced for all the available classes.

**Batch Statistics Loss.** Prior works [22, 57, 50] in the centralized setting have recognized that the distribution of synthetic images generated by model inversion methods can drift from real data. Therefore, in order to avoid such problems, we add batch statistics loss $\mathcal{L}_{BN}$ to our generator training objective. Specifically, the server has access to the statistics (mean and standard deviation) of the global model's BatchNorm layers obtained from training on real data. We want to enforce the same statistics in all BatchNorm layers on the generated synthetic images as well. To this end, we minimize the layer-wise distances between the two statistics written as

$$\mathcal{L}_{BN} = \frac{1}{L} \sum_{i=1}^{L} KL(\mathcal{N}(\mu_i, \sigma_i^2), \mathcal{N}(\tilde{\mu}_i, \tilde{\sigma}_i^2)) = \log \frac{\hat{\sigma}}{\sigma} - \frac{1}{2}(1 - \frac{\sigma^2 + (\mu - \hat{\mu})^2}{\hat{\sigma}^2}). \tag{3}$$

Here, $L$ denotes the total number of BatchNorm layers, $\mu_i$ and $\sigma_i$ are the mean and standard deviation stored in BatchNorm layer $i$ of the global model, $\tilde{\mu}_i$, $\tilde{\sigma}_i$ are measured statistics of BatchNorm layer $i$ for the synthetic images. Finally, $KL$ stands for the Kullback-Leibler (KL) divergence.

We want to note that this loss does not rely on the BatchNorm layer itself but rather on their stored statistics ($\tilde{\mu}_i$, $\tilde{\sigma}_i$ ). $\mathcal{G}$ aims to generate synthetic images similar to the real ones such that the global model would not be able to classify them purely based on these statistics. One way to achieve this is to ensure that synthetic and real images have similar statistics in the intermediate layers, and this is

the role of $\mathcal{L}_{BN}$. In our experiments, we employed the most common baseline model in CL, which already contains BatchNorm layers and measures those statistics. However, these layers are not a necessity and can be substituted by similar ones, such as GroupNorm. In general, if no normalization layer is used in the model, clients can still compute the running statistics of specific layers and share them with the server, and later, the server can use them in the training of the $\mathcal{G}$.

**Image Prior Loss.** In natural images, adjacent pixels usually have values close to each other. Adding prior loss is a common technique to encourage a similar trend in the synthetic images [22]. In particular, we can create the smoothed (blurred) version of an image by applying a Gaussian kernel and minimizing the distance of the original and $Smooth(\tilde{x})$ using the image prior loss

$$\mathcal{L}_{pr} = ||\tilde{x} - Smooth(\tilde{x})||_2^2. \tag{4}$$

In summary, we can write the training objective of $\mathcal{G}$ as Equation 5 where $w_{div}$, $w_{BN}$ and $w_{pr}$ control weight of each term.

$$\min_{\mathcal{G}} \mathcal{L}_{CE} + w_{div}\mathcal{L}_{div} + w_{BN}\mathcal{L}_{BN} + w_{pr}\mathcal{L}_{pr}, \tag{5}$$

### 3.2 Client-side: Continual Learning

For client-side training, our solution is inspired by the algorithm proposed in [50]. In particular, the authors distill the *stability-plasticity* dilemma into three critical requirements of continual learning and aim to address them one by one.

**Current Task.** To have plasticity, the model needs to learn the new features in a way that is least biased towards the old tasks. Therefore, instead of including all the output space in the loss, the CE loss can be computed *for the new classes only* by splitting the linear heads and excluding the old ones, which we can write as

$$\mathcal{L}_{CE}^t = \begin{cases} CE(\mathcal{F}_t(x), y), & if \ y \in \mathcal{Y}^t \\ 0, & O.W. \end{cases} \tag{6}$$

**Previous Tasks.** To overcome forgetting, after the first task, we train the model using synthetic and real data simultaneously. However, the distribution of the synthetic data might differ from the real one, and it becomes important to prevent the model from distinguishing old and new data only based on the distribution difference. To address this problem, we only use the extracted features of the data. To this aim, clients freeze the feature extraction part and only update the classification head (represented by $\mathcal{F}_t^*$) for both real ($x$) and synthetic ($\tilde{x}$) images. This fine-tuning loss is formulated as

$$\mathcal{L}_{FT}^t = CE(\mathcal{F}_t^*([x, \tilde{x}]), y). \tag{7}$$

Finally, to minimize feature drift and forgetting of the previous tasks, the common method is knowledge distillation over the prediction layer. However, [50] proposed *importance-weighted feature distillation*: instead of using the knowledge in the decision layer, they use the output of the feature extraction part of the model (penultimate layer). This way, only the more significant features of the old model are transferred, enabling the model to learn the new features from the new tasks. This loss can be written as

$$\mathcal{L}_{KD}^t = ||\mathcal{W}(\mathcal{F}_t^{1:L-1}([x, \tilde{x}])) - \mathcal{W}(\mathcal{F}_{t-1}^{1:L-1}([x, \tilde{x}]))||_2^2, \tag{8}$$

where $\mathcal{W}$ is the frozen linear head of the model trained on the last task ($\mathcal{W} = \mathcal{F}_{t-1}^L$).
In summary, the final objective on the client side as

$$\min_{\mathcal{F}_t} \mathcal{L}_{CE}^t + w_{FT}\mathcal{L}_{FT}^t + w_{KD}\mathcal{L}_{KD}^t, \tag{9}$$

where $w_{FT}$ and $w_{KD}$ are hyper-parameters determining the importance of each loss term.

### 3.3 Summary of MFCL Algorithm

In summary, during the first task, clients train the model using only the $\mathcal{L}_{CE}$ part of (9) and send their updates to the server where the global model gets updated (FedAvg) for $R$ rounds. At the end of

training task $t = 1$, the server trains the generative model by optimizing (5), using the latest global model. Finally, the server freezes and saves $\mathcal{G}$ and the global model ($\mathcal{F}_{t-1}$). This procedure repeats for all future tasks, with the only difference being that for $t > 1$, the server needs to send the current global model ($\mathcal{F}_t$), precious task's final model ($\mathcal{F}_{t-1}$) and $\mathcal{G}$ to clients. Since $\mathcal{F}_{t-1}$ and $\mathcal{G}$ are fixed during training $\mathcal{F}_t$, the server can send them to each client once per task to reduce the communication cost. To further decrease this overhead, we can employ communication-efficient methods in federated learning, such as [5], that can highly compress the model with minor performance degradation, which we leave for future work. Algorithm 1 in the Appendix A shows different steps of MFCL.

## 4  SuperImageNet

In centralized Class-IL, the tasks are disjoint, and each task reveals a new set of classes; therefore, the total number of classes strongly limits the number of tasks. Moreover, we must ensure that each task has sufficient training data for learning. Thus, the number of examples per class is essential in creating CL datasets. However, the dataset needs to be split along the task dimension and clients in a Federated Class-IL setup. For instance, CIFAR-100, a popular dataset for benchmarking FL algorithms, consists of $100$ classes, each with $500$ examples, which must be partitioned into $T$ tasks, and each task's data is split among $N$ clients. In other words, for a single task, a client has access to only $\frac{1}{T \times N}$ of that dataset; in a common scenario where $N = 100$ and $T = 10$, we can assign only $50$ samples to each client (about 5 example per class in i.i.d data distribution), which is hardly enough.

To resolve this problem, prior works have used a small number of clients [45, 24, 52], combined multiple datasets [58], employed a surrogate dataset [37] or allowed data sharing among the clients [15]. However, these solutions may not be possible, applicable, or may violate the FL's assumptions. This demonstrates the importance of introducing new benchmark datasets for federated continual settings.

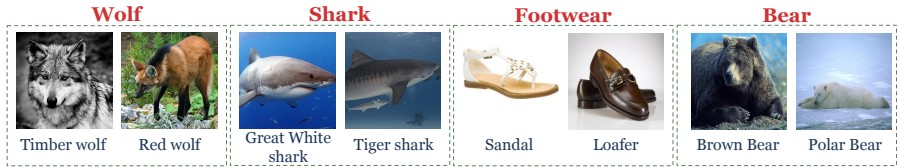

Figure 4: Building SuperImageNet by regrouping ImageNet dataset. Labels in Blue are the original labels, and in Red are the labels in SuperImageNet.

We introduce **SuperImageNet**, a dataset created by superclassing the *ImageNet* [14] dataset, thus greatly increasing the number of available samples for each class. There are 3 versions of the dataset, each offering a different trade-off between the number of classes (for Class-IL) and the number of examples per class (for FL) as shown in Table 4. For example, *SuperImageNet-M* has *10x* more samples per class compared to CIFAR-100, which allows for an order of magnitude increase in the number of fed-

| Dataset | # examples/class | # classes |
|---|---|---|
| *SuperImageNet-S* | 2500 | 100 |
| *SuperImageNet-M* | 5000 | 75 |
| *SuperImageNet-L* | 7500 | 50 |

Table 2: Versions of SuperImageNet

erated clients in while maintaining the same amount of training data per client. As shown in Figure 4, we have merged classes of similar concepts to increase the sample size per class.

## 5  Experiments

**Setting.** We demonstrate the efficacy of our method on three challenging datasets: CIFAR-100 [30], TinyImageNet [43] and SuperImageNet-L [1]. For all datasets, we use the baseline ResNet18 [23] as the global model and ConvNet architecture for $\mathcal{G}$, which we explain in detail in the Appendix C.

---

[1]The image size of the CIFAR-100, TinyImageNet, and SuperImageNet datasets is $32 \times 32$, $64 \times 64$ and $224 \times 224$, respectively

Table 3 summarizes the setting for each dataset. For each dataset, there are 10 non-overlapping tasks ($T = 10$), and we use Latent Dirichlet Allocation (LDA) [46] with $\alpha = 1$ to distribute the data of each task among the clients. Clients train the local model using an SGD optimizer, and all the results were reported after averaging over 3 different random initializations (seeds). We refer to Appendix F for other hyperparameters.

| Dataset | #Client | #Client per round | #classes per task |
|---|---|---|---|
| CIFAR-100 | 50 | 5 | 10 |
| TinyImageNet | 100 | 10 | 20 |
| SuperImageNet-L | 300 | 30 | 5 |

Table 3: Training parameters of each dataset.

**Metric.** We use three metrics –Average Accuracy, Average Forgetting, and Wallclock time.

*Average Accuracy ($\tilde{\mathcal{A}}$)*: Let us define Accuracy ($\mathcal{A}^t$) as the accuracy of the model at the end of task $t$, over *all* the classes observed so far. Then, $\tilde{\mathcal{A}}$ is average of all $\mathcal{A}^t$ for all the $T$ available tasks.

*Average Forgetting ($\tilde{f}$)*: Forgetting ($f^t$) of task $t$ is defined as the difference between the highest accuracy of the model on task $t$ and its performance at the end of the training. Therefore, we can evaluate the average forgetting by averaging all the $f^t$ for task 1 to $T - 1$ at the end of task $T$.

*Wallclock time.* This is the time the server or clients take to perform one FL round in seconds. The time is measured rounds on our local GPU NVIDIA-A100 and averaged between different clients.

**Baseline.** We compare our method with **FedAvg** [40], **FedProx** [33], **FedProx$^+$**, **FedCIL** [45], **FedLwF-2T** [52] and **Oracle**. **FedAvg** and **FedProx** are the two most common aggregation methods; specifically, FedProx is designed for non-i.i.d data distributions and tries to minimize the distance of the client's update from the global model. Inspired by FedProx, we also explore adding a loss term to minimize the change of the current global model from one from the previous task, which we name **FedProx$^+$**. **FedCIL** is a GAN-based method where clients train the discriminator and generator locally to generate synthetic samples from the old tasks. **FedLwF-2T** is another method designed for federated continual learning. In this method, clients have two additional knowledge distillation loss terms: their local model trained on the previous task and the current global model. Finally, **Oracle** is an upper bound on the performance: during the training of the $i_{th}$ task, clients have access to all of their training data from $t = 1$ to $t = i$.

## 5.1 Results

Figure 5 shows the accuracy of the model on all the observed classes so far. In all three datasets, MFCL consistently outperforms the baselines by a large margin (up to $25\%$ absolute improvement in test accuracy). In the CIFAR-100 dataset, the only baseline that can also correctly classify some examples from past data is **FedCIL**. Both MFCL and FedCIL benefit from a generative model (roughly the same size) to remember the past. Here, a similar generative model to the one in the [45] for the CIFAR-10 dataset is used. Since, in FedCIL, the clients train the generative and global models simultaneously, they require more training iteration. We repeat the same process and adapt similar architectures for the other two datasets. [2] But, given that GANs are not straightforward to fine-tune, this method does not perform well or converge. We explain more in the Appendix H.

We have further compared the performance and overhead of the methods in Table 4. The first two metrics, Average Accuracy and Average Forgetting reveal how much the model is learning new tasks while preserving its performance on the old task. As expected, FedAvg and FedProx have the highest forgetting values because they are not designed for such a scenario. Also, high forgetting for FedLwF-2T indicates that including teachers in the absence of old data cannot be effective. Notably, FedProx$^+$ has a lower forgetting value, mainly due to the fact that it also has lower performance for each task. Finally, FedCIL and MFCL have experienced the least forgetting with knowledge transferred from the old task to the new ones. Particularly, MFCL has the smallest forgetting, which means it is the most successful in preserving the learned knowledge.

We also compare the methods based on their computational costs. It is notable that some methods change after learning the first task; therefore, we distinguish between the cost of the first task and the other ones. As depicted, for $T > 1$, MFCL slightly increases the training time caused by employing the generative model. But, as a trade-off, it can significantly improve performance and forgetting.

---

[2]This result might improve by allocating relatively more resources to the clients.

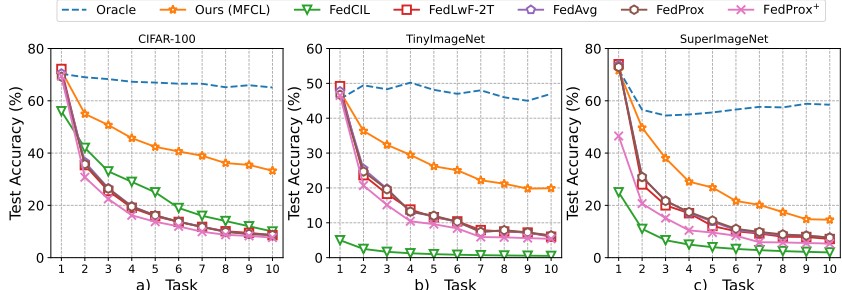

Figure 5: Test Accuracy vs. # observed tasks for (a) CIFAR-100, (b) TinyImageNet, (C) SuperImageNet-L datasets. After each task, the model is evaluated on all the seen tasks so far.

The server cost in MFCL is similar to FedAvg except at the end of each task, where it needs to train the generative model. This extra computation cost should not be a bottleneck because it occurs once per task, and servers usually have access to better computing power compared to clients.

Table 4: Performance of the different baselines in terms of Average Accuracy. Average Forgetting and Wallclock time for CIFAR-100 dataset.

|  | Average Accuracy $\tilde{\mathcal{A}}$ (%) | Average forgetting $\tilde{f}$(%) | Training time (s) $(T = 1)$ | Training time (s) $(T > 1)$ | Server Runtime (s) |
|---|---|---|---|---|---|
| **FedAvg** | $22.27 \pm 0.22$ | $78.77 \pm 0.83$ | $\approx 1.2$ | $\approx 1.2$ | $\approx 1.8$ |
| **FedProx** | $22.00 \pm 0.31$ | $78.17 \pm 0.33$ | $\approx 1.98$ | $\approx 1.98$ | $\approx 1.8$ |
| **FedCIL** | $26.8 \pm 0.44$ | $38.19 \pm 0.31$ | $\approx 17.8$ | $\approx 24.5$ | $\approx 2.5$ for $T = 1$, $\approx 4.55$ for $T > 1$ |
| **FedLwF-2T** | $22.17 \pm 0.13$ | $75.08 \pm 0.72$ | $\approx 1.2$ | $\approx 3.4$ | $\approx 1.8$ |
| **MFCL (Ours)** | $\mathbf{44.98 \pm 0.12}$ | $\mathbf{28.3 \pm 0.78}$ | $\approx 1.2$ | $\approx 3.7$ | $\approx 330$ (once per task), $\approx 1.8$ O.W. |
| **Oracle** | $67.12 \pm 0.4$ | $--$ | $\approx 1.2$ | $\approx 1.2 \times T$ | $\approx 1.8$ |

## 5.2 Ablation Studies

Here, we demonstrate the importance of each component in our proposed algorithm, both on the server and client side, by ablating their effects one by one. Table 5 shows our results, where each row removes a single loss component, and each column represents the corresponding test accuracy ($\mathcal{A}^t$), average accuracy ($\tilde{\mathcal{A}}$), average forgetting ($\tilde{f}$) and their difference from our proposed method. The first three rows are the losses for training the generative model. Our experiments show that Batch Statistics Loss ($\mathcal{L}_{BN}$) and Diversity loss ($\mathcal{L}_{div}$) play an essential role in the final performance. The next three rows reflect the importance of client-side training. In particular, the fourth row (*Ours-w/o $\mathcal{L}_{CE}^t$*) represents the case where clients use all the linear heads of the model for cross-entropy instead of splitting the heads and using the part related to the current task only. The following two rows show the impact of removing $\mathcal{L}_{FT}^t$ and $\mathcal{L}_{KD}^t$ from the client loss. In all three cases, the loss considerably drops, demonstrating the importance of all components. Finally, FedAvg + Gen shows the performance of the case where the server trains the generative model, and clients use its synthetic data the same way as the real ones without further modifications. In the Appendix G, we perform additional ablations on hyperparameters, such as weights of each loss term, generator model size, and noise dimension.

Table 5: Ablation study for MFCL on CIFAR-100

| Method | $\mathcal{A}^1$ | $\mathcal{A}^2$ | $\mathcal{A}^3$ | $\mathcal{A}^4$ | $\mathcal{A}^5$ | $\mathcal{A}^6$ | $\mathcal{A}^7$ | $\mathcal{A}^8$ | $\mathcal{A}^9$ | $\mathcal{A}^{10}$ | $\tilde{\mathcal{A}}$ | $\Delta$ | $\tilde{F}$ | $\Delta$ |
|---|---|---|---|---|---|---|---|---|---|---|---|---|---|---|
| Ours-w/o $\mathcal{L}_{BN}$ | 70.00 | 47.02 | 43.93 | 38.98 | 35.98 | 34.14 | 32.60 | 30.17 | 27.93 | 24.36 | 38.51 | $-6.47$ | 45.95 | $+17.65$ |
| Ours-w/o $\mathcal{L}_{pr}$ | 70.47 | 52.33 | 49.90 | 44.87 | 42.09 | 39.56 | 38.18 | 35.21 | 33.74 | 32.40 | 43.87 | $-1.11$ | 29.47 | $+1.17$ |
| Ours-w/o $\mathcal{L}_{div}$ | 69.87 | 53.48 | 47.60 | 39.60 | 35.43 | 32.95 | 30.81 | 27.15 | 25.14 | 22.34 | 38.44 | $-6.54$ | 44.80 | $+16.5$ |
| Ours-w/o $\mathcal{L}_{CE}^t$ | 70.10 | 40.10 | 33.40 | 26.70 | 21.33 | 19.24 | 17.96 | 14.00 | 13.69 | 11.28 | 26.78 | $-18.20$ | 72.24 | $+43.94$ |
| Ours-w/o $\mathcal{L}_{FT}^t$ | 70.37 | 46.17 | 42.16 | 37.57 | 33.91 | 32.29 | 30.94 | 28.25 | 27.00 | 24.64 | 37.33 | $-7.65$ | 42.85 | $+14.55$ |
| Ours-w/o $\mathcal{L}_{KD}^t$ | 70.10 | 45.92 | 38.60 | 31.01 | 26.45 | 24.07 | 21.32 | 18.02 | 16.85 | 16.29 | 30.86 | $-14.12$ | 53.64 | $+25.34$ |
| FedAvg + Gen | 70.57 | 40.07 | 30.91 | 23.75 | 20.38 | 17.56 | 16.02 | 12.90 | 13.18 | 11.57 | 25.69 | $-19.29$ | 60.46 | $+32.16$ |
| Ours | 71.50 | 55.00 | 50.73 | 45.73 | 42.38 | 40.62 | 38.97 | 36.18 | 35.47 | 33.25 | 44.98 | $-$ | 28.3 | $-$ |

# 6 Discussion

**Privacy of MFCL.** Federated Learning, specifically FedAvg, is vulnerable to different attacks, such as data poisoning, model poisoning, backdoor attacks, and gradient inversion attacks [27, 36, 18, 20, 11, 33]. We believe, MFCL generally does not introduce any additional privacy issues and still it is prone to the same set of attacks as FedAvg. MFCL trains the generative model based on the weights of the *global model*, which is already available to all clients in the case of FedAvg. On the contrary, in some prior work in federated continual learning, the clients need to share a locally trained generative model or perturbed private data, potentially causing more privacy problems.

Furthermore, for FedAvg, various solutions and defenses, such as differential privacy or secure aggregation [55, 8], are proposed to mitigate the effect of such privacy attacks. One can employ these solutions in the case of MFCL as well. Notably, in MFCL, the server **does not** require access to the individual client's updates and uses the aggregated model for training. Therefore, training a generative model is still viable after incorporating these mechanisms.

In MFCL, the server trains the generator using only client updates. Figure 6 presents random samples of real and synthetic images from the CIFAR-100 dataset. Images of the same column correspond to real and synthetic samples from the same class. Synthetic samples do not resemble any specific training examples of the clients and thus preserve privacy. However, they consist of some common knowledge about the class and effectively represent the whole class. Therefore, they can significantly reduce catastrophic forgetting.

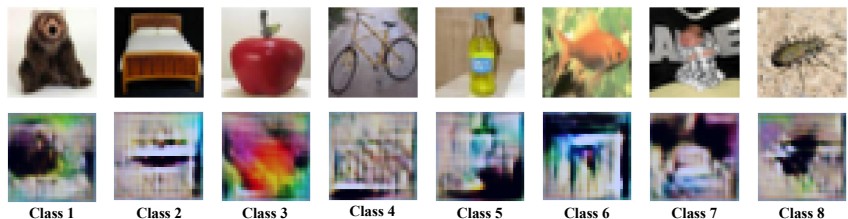

Figure 6: Real vs synthetic data generated by the generative model for CIFAR-100 dataset.

**Limitations.** In our method, clients need the generative model, the final global model of the last task, and the current global model, which adds overheads such as communication between the server and clients and storage. However, there are fundamental differences between storing the generative model and actual data. First, the memory cost is independent of the task size: as the number of tasks increases, clients either have to delete some of the existing examples of the memory to be able to add new ones or need to increase the memory size. In contrast, the generative model size is constant. Finally, clients can delete the generative model while not participating in the FL process and retrieve it later if they join. On the other hand, deleting data samples from memory results in a permanent loss of information. We have delved into this in Appendix D.

# 7 Conclusion

This work presents a federated Class-IL framework while addressing resource limitations and privacy challenges. We exploit generative models trained by the server in a data-free fashion, obviating the need for expensive on-device memory on clients. Our experiments demonstrate that our method can effectively alleviate catastrophic forgetting and outperform the existing state-of-the-art solutions.

# 8 Acknowledgment

This material is based upon work supported by ONR grant N00014-23-1-2191, ARO grant W911NF-22-1-0165, Defense Advanced Research Projects Agency (DARPA) under Contract No. FASTNICS HR001120C0088 and HR001120C0160, and gifts from Intel and Qualcomm. The views, opinions, and/or findings expressed are those of the author(s) and should not be interpreted as representing the official views or policies of the Department of Defense or the U.S. Government.

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

# A MFCL Algorithm

Algorithm 1 summarizes our method. Here, for every task, clients train the local model using the shared generative model. At the end of each task, the server updates the generative model using data-free methods.

---

**Algorithm 1** MFCL

---

1: $N$: #Clients, $[\mathcal{C}_N]$: Client Set, $K$: #Clients per Round, $u_i$: client i Update, $E$: Local Epoch
2: $R$: FL Rounds per Task, $T$: #Tasks, $t$: current task , $|\mathcal{Y}|^t$: Task $t$ Size, $q$: #Discovered Classes
3: $\mathcal{F}_t$: Global Model for task t, $\mathcal{G}_t$: Generative Model, $E_{\mathcal{G}}$: Generator Training Epoch
4: $q \leftarrow 0$
5: $\mathcal{G}, \mathcal{F}_1 \leftarrow$ **initialize**()
6: **for** $t = 1$ **to** $T$ **do**
7:     $q \leftarrow q + |\mathcal{Y}^t|$
8:     $\mathcal{F}_t \leftarrow$ **updateArchitecture**$(\mathcal{F}_t, q)$ # Add new observed classes in the classification layer.
9:     **for** $r = 1$ **to** $R$ **do**
10:       $C_K \leftarrow$ **RandomSelect**$([\mathcal{C}_N], K)$
11:       **for** $c \in C_K$ in parallel **do**
12:         $u_c \leftarrow$ **localUpdate**$(\mathcal{F}_t, \mathcal{G}, \mathcal{F}_{t-1}, E)$ # For $t = 1$ we do not need $\mathcal{F}_0$ and $\mathcal{G}$.
13:       **end for**
14:       $\mathcal{F}_t \leftarrow$ **globalAggregation**$(\mathcal{F}_t, [u_c])$
15:     **end for**
16:     $\mathcal{F}_t \leftarrow$ **freezeModel**$(\mathcal{F}_t)$ # Fix Global model.
17:     $\mathcal{G} \leftarrow$ **trainDFGenerator**$(\mathcal{F}_t, E_{\mathcal{G}}, q)$ # Train the generative model.
18:     $\mathcal{G} \leftarrow$ **freezeModel**$(\mathcal{G})$ # Fix generator weights.
19: **end for**

---

# B Code for Reproduction

The codebase for this work and regrouping the ImageNet dataset is available at `https://github.com/SaraBabakN/MFCL-NeurIPS23`.

# C Details of the Generative Model

**Architectures.** In Table 6, we show the generative model architectures used for CIFAR-100, TinyImageNet, and SuperImageNet datasets. In all experiments, the global model has ResNet18 architecture. For the CIFAR-100 and TinyImageNet datasets, we change the first `CONV` layer kernel size to $3 \times 3$ from $7 \times 7$. In this table, `CONV` layers are reported as $\text{CONV} K \times K(C_{in}, C_{out})$, where $K$, $C_{in}$ and $C_{out}$ are the size of the kernel, input channel and output channel of the layer, respectively.

**Weight Initialization.** The generative model is randomly initialized for the first task and trained from scratch. For all the future tasks (t > 1), the server uses the previous generative model (t - 1) as the initialization.

**Synthetic Samples Generation.** To generate the synthetic data, clients sample i.i.d noise, which later would determine the classes via the argmax function applied to the first q elements (considering q is the total number of seen classes). Given the noise is sampled i.i.d, the probability of generating samples from class $i$ equals $\frac{1}{q}$. Although this might not lead to the same number of synthetic samples from each class in every batch, the generated class distribution is uniform over all classes. Thus, in expectation, we have class balance in generated samples.

**Catastrophic Forgetting in the Generative Model.** The effectiveness of the $\mathcal{G}$ is closely linked to the performance of the global model. If the global model forgets old classes after completing a task, the quality of corresponding synthetic data will decline. Hence, it is crucial to select a reliable generative model and a robust global model. A good generative model can assist the global model in preventing forgetting when learning new tasks. This model can then serve as a teacher for the next round of the $\mathcal{G}$ model.

**Global Aggregation Method.** In this work, we have employed FedAvg to aggregate the client updates. Since the generator is always trained after the aggregation, its training is not impacted by changing the aggregation method. However, the generative model uses the aggregated model as its discriminator, and it is directly affected by the quality of the final global model. Therefore, any aggregation mechanism that improves the global model's performance would also help the generative model and vice versa.

Table 6: Generative model Architecture

| CIFAR-100 | TinyImageNet | SuperImageNet |
|---|---|---|
| $\text{FC}(200, 128 \times 8 \times 8)$ | $\text{FC}(400, 128 \times 8 \times 8)$ | $\text{FC}(200, 64 \times 7 \times 7)$ |
| $\text{reshape}(-, 128, 8, 8)$ | $\text{reshape}(-, 128, 8, 8)$ | $\text{reshape}(-, 64, 7, 7)$ |
| $\text{BatchNorm}(128)$ | $\text{BatchNorm}(128)$ | $\text{BatchNorm}(64)$ |
| $\text{Interpolate}(2)$ | $\text{Interpolate}(2)$ | $\text{Interpolate}(2)$ |
| $\text{CONV3} \times 3(128, 128)$ | $\text{CONV3} \times 3(128, 128)$ | $\text{CONV3} \times 3(64, 64)$ |
| $\text{BatchNorm}(128)$ | $\text{BatchNorm}(128)$ | $\text{BatchNorm}(64)$ |
| $\text{LeakyReLU}$ | $\text{LeakyReLU}$ | $\text{LeakyReLU}$ |
| $\text{Interpolate}(2)$ | $\text{Interpolate}(2)$ | $\text{Interpolate}(2)$ |
| $\text{CONV3} \times 3(128, 64)$ | $\text{CONV3} \times 3(128, 128)$ | $\text{CONV3} \times 3(64, 64)$ |
| $\text{BatchNorm}(64)$ | $\text{BatchNorm}(128)$ | $\text{BatchNorm}(64)$ |
| $\text{LeakyReLU}$ | $\text{LeakyReLU}$ | $\text{LeakyReLU}$ |
| $\text{CONV3} \times 3(64, 3)$ | $\text{Interpolate}(2)$ | $\text{Interpolate}(2)$ |
| $\text{Tanh}$ | $\text{CONV3} \times 3(128, 64)$ | $\text{CONV3} \times 3(64, 64)$ |
| $\text{BatchNorm}(3)$ | $\text{BatchNorm}(3)$ | $\text{BatchNorm}(64)$ |
| | $\text{LeakyReLU}$ | $\text{LeakyReLU}$ |
| | $\text{CONV3} \times 3(64, 3)$ | $\text{Interpolate}(2)$ |
| | $\text{Tanh}$ | $\text{CONV3} \times 3(64, 64)$ |
| | $\text{BatchNorm}(3)$ | $\text{BatchNorm}(64)$ |
| | | $\text{LeakyReLU}$ |
| | | $\text{Interpolate}(2)$ |
| | | $\text{CONV3} \times 3(64, 64)$ |
| | | $\text{BatchNorm}(64)$ |
| | | $\text{LeakyReLU}$ |
| | | $\text{CONV3} \times 3(64, 3)$ |
| | | $\text{Tanh}$ |
| | | $\text{BatchNorm}(3)$ |

# D  Overheads of generative model

**Client-side.** Using $\mathcal{G}$ on the client side would increase the computational costs compared to vanilla FedAvg. However, existing methods in CL often need to impose additional costs such as memory, computing, or both to mitigate catastrophic forgetting. Nevertheless, there are ways to reduce costs for MFCL. For example, clients can perform inference once, generate and store synthetic images only for training, and then delete them all. They can further reduce costs by requesting that the server generate synthetic images and send them the data instead of $\mathcal{G}$. Here, we raise two crucial points about the synthesized data. Firstly, there is an intrinsic distinction between storing synthetic (or $\mathcal{G}$) and actual data; the former is solely required during training, and clients can delete them right after the training. Conversely, the data in episodic memory should always be saved on the client's side because once deleted, it becomes unavailable. Secondly, synthetic data is shared knowledge that can assist anyone with unbalanced data or no memory in enhancing their model's performance. In contrast, episodic memory can only be used by one client.

**Server-side.** The server needs to train the $\mathcal{G}$ **once per task**. It is commonly assumed that the server has access to more powerful computing power and can compute more information in a faster time compared to clients. This training step does not have overhead on the client side and might slow down the whole process. However, tasks do not change rapidly in real life, giving the server ample time to train the generative model before any trends or client data shifts occur.

**Communication Cost.** Transmitting the generative model can be a potential overhead for MFCL, as it is a cost that clients must bear **once per task** to prevent or reduce catastrophic forgetting. However, several possible methods, such as compression, can significantly reduce this cost while maintaining excellent performance. This could be an interesting direction for future research.

# E  More on the Privacy of MFCL

**MFCL with Differential Privacy.** We want to highlight that the generator can only be as good as the discriminator in data-free generative model training. If the global model can learn the decision boundaries and individual classes with a DP guarantee, the generator can learn this knowledge and present it through the synthetic example. Otherwise, if the global model fails to learn the current tasks, there is not much knowledge to preserve for the future. With the DP guarantee, the main challenge is training a reasonable global model; improving this performance can also help the generative model.

**MFCL with Secure Aggregation.** If the clients do not trust the server with their updates, a potential solution is *Secure Aggregation*. In a nutshell, secure aggregation is a defense mechanism that ensures update privacy, especially when the server is potentially malicious. More importantly, since MFCL also does not require individual updates, it is compatible with secure aggregation and can be employed to align with Secure Aggregation.

**Privacy Concerns Associated with Data Storage.** Currently, some different regulations and rules limit the storage time of users' data. Usually, the service providers do not own the data forever and are obligated to erase it after a specific duration. Sometimes, the data is available only in the form of a stream, and it never gets stored. But most of the time, data is available for a short period of enough to perform a few rounds of training. In this way, if multiple service providers participate in federated learning, their data would dynamically change as they delete old data and acquire new ones.

**MFCL and Batch Statistics.** MFCL benefits from Batch Statistics Loss ($\mathcal{L}_{BN}$) in training the generative model. However, some defense mechanisms suggest not sharing local Batch Statistics with the server. While training the generative model without the $\mathcal{L}_{BN}$ is still possible, it can reduce the accuracy. Addressing this is an interesting future direction.

# F  Hyperparameters

Table 7 presents some of the more important parameters and settings for each experiment.

Table 7: Parameter Settings in different datasets

| Dataset | CIFAR-100 | TinyImageNet | SuperImageNet-L |
|---|---|---|---|
| **Data Size** | $32 \times 32$ | $64 \times 64$ | $224 \times 224$ |
| **# Tasks** | 10 | 10 | 10 |
| **# Classes per task** | 10 | 20 | 5 |
| **# Samples per class** | 500 | 500 | 7500 |
| **LR** | All task start with 0.1 and exponentially decay to 0.01 | | |
| **Batch Size** | 32 | 32 | 32 |
| **Synthetic Batch Size** | 32 | 32 | 32 |
| **FL round per task** | 100 | 100 | 100 |
| **Local epoch** | 10 | 10 | 1 |

# G  Hyperparameter tuning for MFCL

Hyperparameters can play an essential role in the final performance of algorithms. In our experiments, we have adapted the commonly used parameters, and here, we show how sensitive the final performance is regarding each hyperparameter. This is particularly important because hyperparameter tuning is very expensive in federated learning and can be unfeasible in continual learning. To this aim, we change one parameter at a time while fixing the rest. In Table 8, we report the final $\tilde{\mathcal{A}}$ of each hyperparameter on CIFAR-100 datasets with 10 tasks.

$w_{div}$: Weight of diversity loss ($\mathcal{L}_{div}$).

$w_{BN}$: Weight of Batch Statistics loss ($\mathcal{L}_{BN}$).

$w_{pr}$: Weight of Image Prior loss ($\mathcal{L}_{FT}$).

$Z\_dim$: Input noise dimension for training the $\mathcal{G}$ model.

$gen\_epoch$: Number of iteration to train the $\mathcal{G}$ model.

This is the setting that we used $w_{div} = 1, w_{BN} = 75, w_{pr} = 0.001, Z\_dim = 200, gen\_epoch = 5000$ and the average accuracy equals $45.1\%$. (There may be a minor difference between this value and the result in the main manuscript. This discrepancy arises because we only ran the ablation for a single seed, whereas the results reported in the primary manuscript are the average of three different seeds.)

Table 8: Effect of different hyperparameters on the final $\tilde{\mathcal{A}}$ (in %) for CIFAR-100 dataset.

| $\mathbf{w_{div}}$ | $\tilde{\mathcal{A}}$ | $\mathbf{w_{BN}}$ | $\tilde{\mathcal{A}}$ | $\mathbf{w_{pr}}$ | $\tilde{\mathcal{A}}$ | $\mathbf{Z\_dim}$ | $\tilde{\mathcal{A}}$ | $\mathbf{gen\_epoch}$ | $\tilde{\mathcal{A}}$ |
|---|---|---|---|---|---|---|---|---|---|
| 0.1 | 44.35 | 0.1 | 40.12 | 0.0001 | 43.10 | 110 | 42.39 | 100 | 40.77 |
| 0.5 | 44.37 | 1 | 43.90 | 0.001 | 45.1 | 200 | 45.1 | 5000 | 45.1 |
| 1 | 45.1 | 10 | 44.77 | 0.01 | 43.56 | 1000 | 45.01 | 10000 | 43.35 |
| 2 | 44.08 | 75 | 45.1 | 0.1 | 44.73 | | | | |
| 5 | 44.57 | 100 | 45.02 | 1 | 44.37 | | | | |

This table shows how robust the final performance is with respect to each parameter, which is preferred both in federated and continual learning problems.

# H  Comparison between MFCL and FedCIL

Here, we would like to highlight some distinctions between our algorithm and FedCIL, both of which aim to alleviate catastrophic forgetting using generative models.

- In FedCIL, **clients** train the local generative model **every round**, which adds great computational overhead. On the other hand, in our approach, the generative model is trained on the server and only **once per task**.

- Training models in GANs usually require a large amount of data that is not commonly available, especially on edge devices. Our data-free generative models address this issue.

- Training the generative model directly from the training dataset may pose a risk of exposing sensitive training data, which contradicts the goal of FL. On the other hand, MFCL uses only the information from the global model.

- FedCIL is limited to simpler datasets and FL settings, such as MNIST and CIFAR10, with fewer clients and less complex architectures. In contrast, our approach can handle more complex datasets, such as CIFAR100, TinyImageNet, and SuperImagenet, with a much larger number of clients.

- Training GAN models usually require more careful hyperparameter tuning. To train FedCIL for TinyImageNet and SuperImageNet, we tried SGD and Adam optimizers with learning rates $\in \{0.1, 0.05, 0.01\}$ and local epoch $\in \{1, 2\}$. Furthermore, we adopt a generative model architecture with a similar input dimension and a total number of parameters in MFCL. However, the model did not converge to a good performance. While a more extensive hyperparameter search might improve the results, it can indicate the difficulty of the hyperparameter tuning of this algorithm. It is worth mentioning that in order to train the CIFAR-10 dataset, we used a local epoch $8\times$ larger than the other baselines; otherwise, the performance on this dataset would also degrade.

In conclusion, FedCIL can be a good fit for a cross-silo federated learning setting with only a few clients, each possessing a large amount of data and computing resources. Meanwhile, while still applicable in the above setting, our method is also suitable for edge devices with limited data and power.

