# OpenReview forum: "A Data-Free Approach to Mitigate Catastrophic Forgetting in Federated Class Incremental Learning for Vision Tasks"
_NeurIPS.cc/2023/Conference — NeurIPS 2023 poster_

### Official Review · Reviewer_zVmx · 2023-07-05

**Soundness:** 3 good
**Presentation:** 4 excellent
**Contribution:** 3 good
**Rating:** 5
**Confidence:** 4

**Summary:**

In this paper, authors propose a new approach to perform class-incremental learning in federated setting. Their approach uses a generative model trained at the server side which generates synthetic images to be used as a replacement for data corresponding to old classes. The authors claim through empirical results that their approach outperform all the earlier approaches for this problem.

**Strengths:**

This paper clearly articulates the class incremental training in a federated learning setting and the related challenges. The paper also clearly explains different loss functions used for training the generative model and the client training paradigms used in the proposed approach. Ablation study is also performed to break down the gains into different constituents of the approach.

**Weaknesses:**



On page 3, authors mention that the main difference between FedCIL and their approach is that generative model in the proposed approach is trained in a data free manner which can reduce client's training time and computation. It's not clear how does this difference leads to such better performance in the experimental results. It will be nice to have some discussion and experiments to explain.

As mentioned on page 4, line 177, the generative model produces samples resembling the original training inputs and given that this model is transmitted to all the clients, why this is not a privacy issue? It will be good to analyze any potential privacy leakage through the shared generative model.

It's not clear the SuperImageNet dataset claimed by the authors, is it just a regrouping of the ImageNet Dataset or something more substantial.

In Fig. 5, it's quite surprising to see the such a poor performance of FedCIL compared to FedAvg and FedProx which are not even designed for this problem. There is not much discussion in the paper on this. Specifically, are these results over one run or multiple runs? What hyper parameter thing has been performed for these approaches.

In Fig. 5, performance of MFCL is falling the most as we go from CIFAR-100 to tinyImageNet to SuperImageNet although the number of samples per task is increasing. It will be good to understand why is this observed?

**Questions:**

Please see above.

---

> ### Author Rebuttal · Authors · 2023-08-09
>
> We thank the reviewer for reading the details of our paper and appreciate their comments. We are grateful for their insightful questions and try to address their concern in detail:
>
> ---
> > FedCIL compared to MFCL and other methods
>
> We have briefly discussed the differences between FedCIL and MFCL and our hyperparameters in section 7 of the appendix.
>
> FedCIL is a GAN-based method where all the participating clients train the discriminative model to align with their local generative model.  In the original paper, the authors show successful results for smaller datasets (MNIST and CIFAR10) and a small number of clients (=5). We believe by increasing the dataset’s difficulty and reducing the number of examples per client; this method would face the following challenges;
> 1. GAN models require training on a large amount of data for a long time. However, in scenarios like CIFAR100 with 50 clients, each client has a few examples of each class.
>
> 2. GAN are commonly known to be very sensitive to hyperparameters, especially in a decentralized setup.
>
> **Hyperparameters.** To train FedCIL for TinyImageNet and SuperImageNet, we tried SGD and Adam optimizers with learning rates $\in${0.1, 0.05, 0.01} and local epoch $\in$ {1, 2}. For a fair comparison, we adopt a generative model architecture with the same input dimension and a total number of parameters as MFCL. Unfortunately, the model did not converge to good performance with the mentioned hyperparameters.
>
> We want to highlight that training FedCIL is substantially more expensive compared to other methods, as individual clients have their local generative model. Although other hyperparameters may improve the results, our evaluation indicates the difficulty of the hyperparameter tuning for FedCIL. It is worth mentioning that in order to train the CIFAR-100 dataset, we used a local epoch $8 \times$ larger than the other baselines; otherwise, the performance on this dataset would also degrade.
>
> Having this said, we believe that FedCIL is designed for cross-silo settings where each client has a lot of computational power and training data. On the contrary, MFCL works well across both cross-silo and cross-device settings.
>
> > Privacy implications of sharing the generative model
>
> In our method, the server relies only on the aggregated global model to train the generative model. Since this global model is already shared with the clients in any standard FL framework (such as FedAvg), clients can anyway train such generative models locally and synthesize data. As a result, we believe our method generally does not introduce any *additional privacy* issues.
>
> It is still possible to conduct other attacks, such as model/data poisoning, that are common in FedAvg. Our claim about privacy is compared to other federated continual learning methods where the clients need to share a locally trained generative model or perturbed private data, causing more privacy problems.
>
> It is noteworthy that common defense mechanisms for FedAvg, such as secure aggregation, are also applicable to our algorithm. In MFCL, the server does not require access to the individual client's updates and uses the aggregated model for training. Therefore, training a generative model is still viable after incorporating these defense mechanisms.
>
> > Performance of MFCL from CIFAR-100 to tinyImageNet to SuperImageNet.
>
> We thank the reviewer for their observation. We believe the main reason behind this performance degradation is the difficulty of the datasets. SuperImagenet resolution is 3 * 224 * 244, while this value is 3 * 32 * 32 for CIFAR100. This difference in quality causes two types of problems;
>
>  1- Training a model on SuperImageNet is more difficult. Even in the centralized setting with all the data present (no FL or CL), the performance of ResNet18 is better on CIFAR100.
>
> 2- As the data resolution increases, generating high-quality synthetic data becomes more challenging.
>
> To address the first problem, clients need to train a larger discriminative model and to resolve the second issue, we should increase the generative model size. In our experiments, we employed the common backbone architecture for the global model, which is more suitable for cross-device FL.
>
> > SuperImageNet dataset
>
> SuperImageNet is a subgroup and regrouping of the ImageNet dataset. We will clear this in the main manuscript to avoid confusion.
>
> ---
> We sincerely thank the reviewer for their time, valuable feedback, and questions. We hope to have addressed their concerns and look forward to engaging in further discussion. If the reviewer finds our response satisfactory, we kindly ask them to revisit their evaluation.

---

> > ### Comment · Reviewer_zVmx · 2023-08-15
> > **Updated rating**
> >
> > I have updated my rating based on the responses above by authors. Thanks!

---

> > > ### Author Response · Authors · 2023-08-17
> > > **Thanks reviewer zVmx**
> > >
> > > We appreciate the reviewer for their constructive questions/comments, finding our rebuttal satisfactory, and raising their rating. We would like to kindly ask if there are any remaining concerns from the initial review so that we might have the opportunity to address those. Once again, thanks for the time to review our work.

---

### Official Review · Reviewer_NkkA · 2023-07-07

**Soundness:** 3 good
**Presentation:** 3 good
**Contribution:** 3 good
**Rating:** 5
**Confidence:** 4

**Summary:**

This paper introduces a framework for federated class incremental learning, which employs a generative model trained at the server. This model is then utilized by the client to generate samples from previous distributions to mitigate catastrophic forgetting.

**Strengths:**

This paper addresses the challenging problem of CIL (Class Incremental Learning) within the FL (Federated Learning) framework. It introduces a novel approach of data-free generative model training in the FL framework. Furthermore, the paper proposes a new benchmark dataset for FCIL called SuperImageNet. The effectiveness of the proposed method is demonstrated through extensive experiments.

**Weaknesses:**

This paper uses small-scale model and generative model. It is uncertain how the communication cost and local computational cost will be affected when scaling up and using a generative model.

**Questions:**

1) Is it practical for all clients to share same task transitions at a specific point in time?
2) If storing data locally becomes an issue, the local model update should be performed in a single iteration, and the samples should be discarded afterward. Allowing multiple iterations implies that data has already been stored locally, thus raising privacy concerns associated with data storage. If limited memory resources are the problem, it is understandable. However, there is a concern about whether the generative model will maintain affordable computational complexity as the scale increases.
3) If the server maliciously trains the data-free generative model using a specific local model instead of the aggregated model, wouldn't this lead to privacy issues?
4) Is the L_KD in Eq. (8) and Eq. (9) same with the L_KL in Table 5?
5) At each task boundary, how did you validate the trained generative model?


**Limitations:**

I have some concerns about the communication cost and local computational cost that may arise when using a generative model at scale. It remains uncertain whether these costs will be negligible still.

---

> ### Author Rebuttal · Authors · 2023-08-09
>
> We thank the reviewer for reading the details of our paper and appreciate their comments highlighting the novelty of our method and our extensive empirical results as the strengths of the paper. We are grateful for their insightful questions and try to address their concern in detail:
>
> ---
> > Practicality of sharing the same task transitions.
>
> We want to highlight that in the majority of existing FL studies, client data remains static throughout the entire process. We believe our paper and, in general, federated continual learning is a step towards making FL more realistic where we can observe the impact of change in local data. Undoubtedly, the next step is to relax the setting and let the users’ data change anytime.
>
> In addition, tasks in centralized continual learning are usually isolated and do not overlap to ensure we only observe the challenges of forgetting. Following the same standard practice, we also assume that clients change their training data at specific times.
>
> In practice, clients and the server can establish an agreement on task transitions to improve the global model’s performance. The server can handle task transitions in this setup, and clients can change their local tasks according to the agreement. For example, consider multiple hospitals training a shared model using patient data. The server can analyze the historical data to identify trends, determine specific timestamps, and ask the clients to train the model on the most common illnesses in the seasons.
>
> > Effect of generative model scale on communication and computation costs.
>
> In general, mitigating forgetting requires additional mechanisms on top of traditional FL and introduces new costs in terms of communication, computation, or memory.
>
> **Comparison with the cost of prior work**
>
> *MFCL scales better with the number of tasks*. In memory-based techniques, clients need to assign a fixed memory to all the tasks (which requires deleting old data as the number of tasks increases and, therefore, degrades performance) or a fixed memory per task (the total memory increases linearly by tasks). In contrast, the generative model size is independent of the number of tasks.
>
> *MFCL clients can delete the model*. The generative model is solely required during training, and clients can delete them afterward. Conversely, memory-based techniques require clients to save the data at all times because once deleted, that data become unavailable.
>
> **Potential ways to reduce the costs of generative models**
>
> There are different design choices in MFCL that one can exploit to trade off these costs. To name a few;
>
> *Computation cost.* MFCL, in its current form, requires clients to generate synthetic data in every batch. Clients can reduce this cost by generating and storing synthetic images once (only for training) and deleting them afterward. To further reduce the cost, clients can request the server to generate and send them synthetic images instead of the generative model.
>
> *Communication cost.* The discussed methods can also reduce communication costs. Another alternative is to send the generative model once per task and store it on the client side. Existing techniques, such as pruning, can further reduce the model size while preserving performance.
>
> *Resource-limited clients*. In terms of scaling the generative model, if there are resource-limited clients among the participants, the server can train different generative models with different qualities. Clients can then select the proper generative model based on their restrictions.
>
> > Privacy concerns associated with data storage.
>
> There are different regulations and rules that limit the storage time of users’ data. Service providers are obligated to erase the data eventually after a specific period. Sometimes, the data is available only in the form of a stream and never gets stored; then, we can only use it once for training. However, most of the time, the regulations allow the service provider to store the data for longer, which is enough to perform a few training rounds. After this period, the local data will change, but our method helps the model preserve its knowledge of the unavailable data from old tasks.
>
> > Malicious server
>
> This is a great question, and we thank the reviewer for pointing this out.
>
> FedAvg and other common aggregation methods where the server has access to clients’ updates are susceptible to such attacks. The crucial point is that MFCL trains the generative model based on the aggregated model, which is already available to all clients and servers in the case of FedAvg. So, MFCL is vulnerable to the same set of attacks as FedAvg (including the server accessing clients’ updates) but does not introduce additional problems. In contrast, prior work requires clients to share a locally trained generative model or perturbed private data, potentially causing more privacy issues.
>
>  **Secure aggregation.** A potential solution when clients do not trust the server with their updates is “Secure Aggregation”. In summary, secure aggregation is a defense mechanism that protects update privacy from malicious servers. Since MFCL does not require individual updates, it is compatible with secure aggregation and can be employed to align with this dense.
>
> > How to validate the generative model.
>
> We validate the generative model in different ways:
>
> 1- Monitoring the trends in each loss term to verify if they are decreasing.
>
> 2- Examining the quality of the synthetic data.
>
> 3- Evaluating the final performance of the global model.
>
> > Are L_KD in Eq. 8, 9 and L_KL in Table 5 equal?
>
> This is a typo, and we will fix it in the paper.
>
> ---
> We sincerely thank the reviewer for their time, valuable feedback, and questions. We hope to have addressed their concerns and look forward to engaging in further discussion. If the reviewer finds our response satisfactory, we kindly ask them to revisit their evaluation.

---

> > ### Comment · Area_Chair_Bkb2 · 2023-08-20
> >
> > Dear authors,
> >
> > Please feel free to add further clarification to reviewer NkkA's comments/questions.
> >
> > Best,
> > AC

---

> > > ### Author Response · Authors · 2023-08-21
> > > **Thanks Area Chair Bkb2 and reviewer NkkA**
> > >
> > > We appreciate the AC for following up on our rebuttal and thank the reviewer for their helpful reviews. However, since we have not received any response from reviewer NkkA on our rebuttal, we are not sure which of their concerns remain and require additional clarification. It would be greatly appreciated if the reviewer could kindly let us know if our rebuttal has properly addressed their concern.
> > >
> > > Here, we provide a summary of our rebuttal;
> > >
> > > 1. **Communication and computation costs at scale**. The extra cost of CL methods is to help clients mitigate forgetting. In our rebuttal, we explained generative models (as we use in our work) scale better than memory-based techniques since (1) the generative model's size is independent of the number of tasks and (2) clients can delete the model after training. In addition, we provided a few ways to reduce the extra overhead in communication and computation.
> > >
> > > 2. **Privacy concerns associated with data storage**.
> > > Data retention policies, in general, are concerned with the purpose and the duration of storing personal data. For example, one of the principles of GDPR (General Data Protection Regulation) requires data to be removed after it is processed for its stated purpose. The permitted storage duration can vary based on the data type and business. However, this duration can be long enough to train the local model for a few iterations.
> > >
> > > 3. **Malicious server**.
> > > A common solution to the mentioned malicious server attack is secure aggregation. We designed MFCL to be compatible with secure aggregation and other common defense mechanisms. The key is that MFCL does not require access to individual client updates and uses the aggregated model for training. Therefore, training the generative model is still viable after incorporating these defense mechanisms.
> > >
> > > 4. **Sharing the same task transitions**. We share this property with recent federated continual learning papers (e.g., FedCIL[1] @ ICLR'23) and follow the standard practice in centralized continual learning to measure the effectiveness of our approach. We believe this setting is closer to practice than conventional FL studies, where the training data does not change. In addition, clients and the server can form an agreement on the transition.
> > >
> > > [1] D. Qi, et al. "Better generative replay for continual federated learning." ICLR, 2023.

---

### Official Review · Reviewer_p34V · 2023-07-08

**Soundness:** 3 good
**Presentation:** 3 good
**Contribution:** 2 fair
**Rating:** 6
**Confidence:** 5

**Summary:**

The work proposes using a server-learned generator for synthetic data replay for federated Class-IL. The method saves client-level compute while preserving data-privacy. The authors show their method outperforms existing methods on 2 existing benchmarks, plus a proposed larger-scale ImageNet benchmark protocol.

**Strengths:**

1) I am already familiar with this setting, but I feel that the authors did a good job at selling the problem-setting to me. While potentially space inefficient, table 1 is very concise and convincing.
2) Moving from multiple client generators -> server generator is very justified, as it will save client compute time. Also, the generator does not need to be communicated from the client to the server, only from the server to the client.
3) The proposed approach does have significantly better performance, especially on the first two datasets.
4) Thank you for the transparency on training costs and server overhead. I do not think the large server overhead is a big deal - since the client costs are minimally affected in T = 1 compared to FedLwF.

**Weaknesses:**

1) The novelty seems to be in more of a high-level idea combination of [45] and federated learning. I am not sure the method is truly impactful from a novelty perspective.
2) SuperImageNet is a protocol benchmark, not a "new benchmark dataset". However, I do appreciate the protocol and agree it is better than CIFAR-100 and TinyImageNet.
3) Overall, the performance is very weak of all methods. I wonder if the impact might be increased with a more realistic setting where past class examples may re-appear in future tasks. Could also consider a small replay buffer and/or a pre-trained model.
4) Often, federated learning papers include some type of theoretical analysis on, for example, convergence.

**Questions:**

a) What specifically would you state are your method contributions compared to a federated variant of [45] and similar approaches?
b) How do you think your method would perform in more realistic federated CL settings where classes are not
Overall, I am worried about the contributed novelty and impact for a venue such as NeurIPS. However, I do think the paper is sound, and in-line with similar works accepted to recent high-tier conferences (e.g., [A]). I am very borderline, but would rather lean towards the accept side.

A. Qi, Daiqing, Handong Zhao, and Sheng Li. "Better generative replay for continual federated learning." arXiv preprint arXiv:2302.13001 (2023).

Other
-> I am not sure that data-free image generation actually protects data-privacy concerns, since you are creating synthetic training data, but, since this line of work is already established, I do not feel the authors need to further justify it.
-> Line 48, might be missing a space between "replay" and "[42]"

**Limitations:**

The discussion on limitations is concise and complete.

---

> ### Author Rebuttal · Authors · 2023-08-10
>
> We thank the reviewer for reading the details of our paper and appreciate their comments highlighting our design choices, superior performance, clarity, and writing as the strengths of the paper. We are grateful for their insightful questions and try to address their concern in detail:
>
> ---
>
> > Novelty of our method
>
> In this paper, we identified the main challenges toward federated continual learning and designed a method to address these challenges. Our framework is **unique** in that it achieves higher accuracy and lower forgetting without requiring any episodic memory and without introducing additional privacy issues compared to the existing alternatives. Our experiment setup is arguably more complex (larger number of clients and larger datasets) than those explored in prior work, such as FedCIL, indicating the effectiveness of our approach.
>
> Even though we borrowed ideas from continual learning (including standard data-free knowledge distillation training and the loss function that has been proven to be effective), we are the first to employ these ideas in the context of federated continual learning. We hope the reviewer agrees that our framework is not a straightforward combination of federated learning and continual learning. We could not outperform other state-of-the-art baselines without our novel design choices, such as deciding to train the generative model on the server (which was pointed out by the reviewer as one of the strengths).
>
>
> > Performance in realistic scenarios.
>
> Please refer to Figure 2 in the attached pdf for the new experiment results.
>
> We define memory as the number of samples from previous tasks. In the paper, we showed the results for $\alpha=1$ and memory size = 0 (meaning that no sample reappears). In the new experiments, we evaluated three different scenarios for the CIFAR100 dataset with 50 clients and evaluated each scenario for two different memory sizes (20 and 50) – each client has 100 samples for each task, so memory size = 50 means 33% of training data is from previous tasks. To pick the reappearing samples, we choose the same number of samples from each previous task and pick the specific sample uniformly at random.
>
> **Scenarios**
>
> A) non-IID data distribution with $\alpha=1$
>
> B) Highly non-IID data distribution with $\alpha=0.1$
>
> C) Highly non-IID data distribution with $\alpha=0.1$ and dynamic client participation. In this scenario, at the end of each task, 5 clients would leave the training, and 5 new clients would join.
>
> **Results.** In all the experiments increasing memory size would improve the final performance. We still see that combining the power of generative models with memory examples has superior performance compared to other baselines.
>
> Due to time constraints, we ran the experiments only on one seed.
>
>
> > Theoretical analysis on convergence.
>
> In this work, we focus on presenting a more realistic experimental setting with various datasets. However, we believe the convergence analysis of our algorithm is mainly determined by the aggregation mechanism.
>
> In our paper, we use FedAvg as the aggregation mechanism in the server, and the convergence of this method is already proved. In FedAvg, the most common local objective function is Cross Entropy (CE). In our algorithm, after task 1, clients add a new term to their objective: $L_{FT}$ and $L_{KD}$. $L_{FT}$ is also a CE loss for synthetic data, and $L_{KD}$ is an MSE loss. Since MSE is L-smooth and $\mu-convex$, we believe adding these two losses does not change the convex property of the original CE loss.
>
> It is also commonly assumed that variance and expected squared norm of stochastic gradients of all the clients are bounded.
>
> Given that adding MSE loss should not change these properties, we believe we can follow the same proof as [1,2] (or similar analysis) for convergence. However, we leave detailed and formal proof for future work.
>
> [1] Li, Xiang, et al. "On the Convergence of FedAvg on Non-IID Data." ICLR. 2019.
>
> [2] Charles, Zachary, and Jakub Konečný. "Convergence and accuracy trade-offs in federated learning and meta-learning." International Conference on Artificial Intelligence and Statistics. PMLR, 2021.
>
> > SuperImageNet is a protocol benchmark.
>
> We agree with the reviewer and will clarify this in the main manuscript.
>
> ---
> We sincerely thank the reviewer for their time, valuable feedback, and questions. We hope to have addressed their concerns and look forward to engaging in further discussion. If the reviewer finds our response satisfactory, we kindly ask them to revisit their evaluation.

---

> > ### Comment · Reviewer_p34V · 2023-08-15
> > **Reviewer p34V response to rebuttal**
> >
> > Due to the hard work by the authors in answering my questions (including the new experiments), I have increased my score to weak accept. I still have some minor reservations on novelty/impact, but the paper is very sound with clear contributions and thus I recommend its acceptance.

---

> > > ### Author Response · Authors · 2023-08-18
> > > **Thanks reviewer p34V**
> > >
> > > We thank the reviewer for their valuable questions/comments, finding our rebuttal satisfactory, and raising their score.

---

### Official Review · Reviewer_4Enn · 2023-07-26

**Soundness:** 3 good
**Presentation:** 2 fair
**Contribution:** 3 good
**Rating:** 6
**Confidence:** 4

**Summary:**

This work introduces MFCL, a method to primarily alleviate the catastrophic forgetting problem that arises in (more realistic) FL settings framed under the continual learning paradigm. In MFCL, the model is split into a generator (only trained on the server side) and a discriminator (only trained by clients). The former is data-free generative model that is trained to output synthetic samples suitable for the task(s) the discriminator is trained (in federation) to perform. This generator is also used by the clients to inject some synthetic training examples into their own local training stages. This component helps counteracting the forgetting problem. The empirical evaluation suggest that MFCL is far superior to other methods.

**Strengths:**

The main strength of this paper is that it studies a far more realistic setting in which FL is used in practice for cross-device settings: clients come and go; data in the clients changes over time (more is added, some might be deleted); and, there is no centralised dataset available on the server. To this problem, MFCL proposes a solution that, although borrowing components and ideas from existing works, have been adapted to the FL setting and they work well.

Other strengths that I have identified:

* The generative model is of a reasonable size (<1M params if my understanding is correct when looking into Table 1 in the Appendix)
* The Authors proposed SuperImageNet dataset.

**Weaknesses:**

The main weaknesses I see in this work are related to lack of clarity in some important point:

* In my view, folks in the FL community might not be familiar with the concept of "task" (common in the CL literature). This means that Sec 3.2 and other parts before might not be so clear. Later in Sec 4 (in line 261 onwards) a clear example of what a _task_ is is presented. Maybe it's worth giving an informal definition earlier?
* The main results (i.e. Figure 5) are not super clear how to interpret. What does it mean "shows the accuracy of the model on the observed classes so far" (line 315)? Is the plot generated at `t=10`  (i.e. after all tasks are completed)?
* Following the comment above: wouldn't it be more informative to have "task" on the x-axis and show how test accuracy (or forgetting) changes as time passes? Presenting the results in such way are more common in some recent CL works I've been reading [1] (see for instance Figures 3,4,5 -- no, I'm not an author :) )
* Figure 3: I think it could be improved adding more details and nomenclature introduced in the text otherwise, what toes the right hand side of Fig3.a tell us that we don't know about all generative training schemes out there? Also, is the "aggregate" green box in Fig3.a outputting the discriminator (yellow rhomboid on the right)? (I think so, so how about connecting them?)


[1] https://arxiv.org/abs/2303.11165

**Questions:**

In addition to the questions I ask in the _Weaknesses_ section, I have a few more:

* Maybe include in the Appendix some additional information about the experimental setup: it seems you used PyTorch, as for FL framework I see there are some utility functions from Flower but for the rest I looks you implemented a custom for loop.
* Why not going a step further with `SuperImageNet` and also fix the number of clients it can be divided into? In FL papers are often hard to compare because people use datasets (synthetic ones specially) in vastly different ways. In my opinion if you were to fix how many partitions it contains, it would help easing the reproducibility problem of FL and ensure subsequent papers that use `SyperImageNet` do so under a common setup. What do you think?
* The role of injecting synthetic images during training (on the client side) could be seen as some form of "alignment" mechanism. This has been presented before in FL [1]. Maybe the Authors could comment whether calling this "alignment" is correct and put it in context with other works? (I only suggested the one that comes to mind immediately, but there are more)
* (very minor style comment) wouldn't It be better to have only full page wide figures/tables (like Fig3) instead of half-page ones (like Tab2,3 and Fig 4) ?

[1] https://arxiv.org/abs/2202.05963

**Limitations:**

There are a few limitations that come to mind after reading this paper:

* How does this method extend to other domains beyond image classification? or image-based problems in general? (I'm inclined to suggest the title of this paper to include the "image" or "vision" keyword)
* In the Discussion, the Authors briefly talk about the privacy implications. I agree that synthetic images do not "resemble any specific training example" but this doesn't have to hold to have some privacy leak. As stated in line 61, the generative model "does not cause any extra data leakage from them [the clients]". But there is a data leak on the discriminator part (just like any other FL method that doesn't apply Differentiable Privacy or similar). Could the Authors comment what would be the implications of adding DP to the discriminator on the client side?

---

> ### Author Rebuttal · Authors · 2023-08-09
>
> We thank the reviewer for reading the details of our paper and appreciate their comments highlighting the strengths of our method and finding the setting more realistic and practical. We are grateful for their insightful questions/suggestions and try to address their concerns in detail;
>
> ---
> > Implications of adding DP to the discriminator.
>
> This is a very interesting question, and we thank the reviewer for raising this point. We want to highlight that in data-free generative model training, the generator can only be as good as the discriminator. If the global model can learn the decision boundaries and individual classes despite having DP, the generator can learn this knowledge and present it through the synthetic example. Otherwise, if the global model fails to learn the current tasks, there is not much knowledge to preserve for the future.
>
> With the DP guarantee, the main challenge is training a reasonable global model. Prior work explores the impact of DP in training the generative model in centralized settings. As an example, [1] shows promising results by training a discriminator with a DP guarantee and training an effective generative model on top of that.
>
> [1] Liu, Bochao, et al. "Privacy-Preserving Student Learning with Differentially Private Data-Free Distillation." 2022 IEEE 24th International Workshop on Multimedia Signal Processing.
>
> > Relation between MFCL and alignment mechanism.
>
> We believe these two lines of work, despite sharing similarities, are mostly orthogonal to each other. Alignment refers to the process of finding a proper permutation of the updates to improve the performance of the aggregated model, whereas in our case, we use a generative model to mitigate forgetting previous tasks. At a high level, both techniques can benefit from shared synthetic data on the client side. However, finding the proper alignment to aggregate does not address the forgetting problem. Our method helps clients to generate data from old tasks (that they may not have access to the data anymore) and inject those samples into their training loop.
>
> We may not have understood the question correctly, and appreciate it if the reviewer could correct us if there is a misunderstanding.
>
> > Clarifying Figures.
>
> **Figure 5.** In Figure 5, we iteratively introduce new tasks. Each task consists of 10, 20, and 5 new classes for CIFAR100, TinyImageNet, and SuperImageNet-L, respectively. To be more specific, the x ticks in all three figures are equivalent to task = {2, 4, 6, 8, 10}. Each point in the plot measures the accuracy of all the classes from the beginning up to that point. For example, the second point from the left shows the accuracy on the first 2 tasks (i.e., 20 classes for CIFAR100). We will clarify this in the paper and will change the x-axis of the figure to show tasks.
>
> **Figure 3.** The right-hand side of figure3(a) emphasizes the difference in frequencies of aggregation and training of the generative model. This is important because the latter is potentially more expensive than aggregation, but the server performs it once per task.
>
> The output of the aggregation is the global model, which we later use as the discriminator in training the generative model. We will replace the figure in the paper with Fig1 in the attached PDF to clarify this.
>
> > Extention to beyond image classification; add "image" or "vision" in the title.
>
> Our method relies on training generative models in a data-free manner. Most prior work in this domain focus on vision tasks since images are more resilient to noise. The generated data may not resemble any meaningful entity but still can help to distinguish different classes. Unfortunately, this may not hold for NLP tasks. As a result, the extension of our method to text data is not straightforward. We will definitely incorporate the reviewer’s suggestion, add ‘image’ or ‘vision’ to the title, and discuss this in the paper.
>
> > Task definition.
>
> We will add the following definition in section 2.
>
> Task: in class incremental learning, the training data is presented to the learner as a sequence of datasets - commonly known as tasks. In each timestamp, only one dataset (task) is available, and the learner’s goal is to perform well on all the current and previous tasks.
>
> > Additional information about the setup.
>
> Our implementation is based on Pytorch 1.13.1 and does not contain any FL framework in the main body. We use FLOWER only for distributing data among clients using LDA.
>
> > SuperImageNet with a fixed number of clients.
>
> We appreciate the reviewer’s suggestion and will release the dataset with the mentioned property.
>
> ---
> We sincerely thank the reviewer for their time, valuable feedback, and questions. We hope to have addressed their concerns and look forward to engaging in further discussion. If the reviewer finds our response satisfactory, we kindly ask them to revisit their evaluation.

---

### Author Rebuttal · Authors · 2023-08-10

We thank all the reviewers for their insights and comments on the paper. Here we have attached a PDF that includes the following;

* A more clarified version of Figure 3.(a) of the paper for reviewer 4Enn.

* More experiment results with increased memory sizes for reviewer p34V.

---

> ### Author Response · Authors · 2023-08-21
> **Thank the reviewers for their insightful feedback**
>
> We are grateful to all the reviewers for reading our paper and for their valuable comments/questions, which helped us improve our manuscript.
>
> We thank Reviewer 4Enn for finding our setting more realistic, identifying our effort in addressing the challenges of adapting continual learning for FL, and highlighting the size of the generative model and SuperImageNet dataset as our strengths; Reviewer p34V for praising our design choices, superior performance, clarity, and writing; Reviewer NkkA for complementing the novelty of our approach, solving a challenging problem, introducing SuperImageNet, and providing extensive empirical results; Reviewer zVmx for praising our empirical results and clarity of writing.
>
> We have provided detailed responses to comments/questions and will incorporate their suggestion in our final manuscript. In the following, we would also like to explain more about the **Privacy** in our method.
>
> 1- Training the generative model is already possible for the server and participants in the existing FL frameworks, such as FedAvg, because this training only requires access to the global model. Therefore, sharing the generative model in our method does not introduce additional privacy issues.
>
> 2- In our method, the server only requires the global weights, not the individual updates. Therefore, our method is compatible with the existing defense mechanisms that protect the privacy of participants' updates.
>
> 3- Our method is more private than the state-of-the-art solutions for federated class incremental learning because it does not require sharing perturbed training data or training the generative model on the local sensitive data.
>
> Thank you again for reviewing our paper and for the insightful comments.

---

### Decision · Program_Chairs · 2023-09-21

**Decision:**

Accept (poster)

**Comment:**

This paper explores an interesting (and realistic, as noted by a reviewer) setting for federated continual learning. The overall approach is an adaptation of existing techniques in CL (i.e., leverage generated data for classes/tasks seen in the past) to the case of federated CL. This is not a trivial task, and the paper does a good job of explaining, justifying, and evaluating the proposed approach. The proposal of a new protocol benchmark is another big plus for the paper. On the downside, reviewer NkkA points to a potential issue with the generator (model selection issues).

In summary, during the discussion phase, the several positive aspects of the paper were noted by the reviewers and the AC. There is also a strong support from several reviewers to accept the paper. On balance, the AC recommends the paper for acceptance.